# Segmented Recurrent Transformer: An Efficient Sequence-to-Sequence Model

**Yinghan Long**
Purdue University
long273@purdue.edu

**Sayeed Shafayet Chowdhury**
Purdue University
chowdh23@purdue.edu

**Kaushik Roy**
Purdue University
kaushik@purdue.edu

## Abstract

Transformers have shown dominant performance across a range of domains including language and vision. However, their computational cost grows quadratically with the sequence length, making their usage prohibitive for resource-constrained applications. To counter this, our approach is to divide the whole sequence into segments and apply attention to the individual segments. We propose a segmented recurrent transformer (SRformer) that combines segmented (local) attention with recurrent attention. The loss caused by reducing the attention window length is compensated by aggregating information across segments with recurrent attention. SRformer leverages Recurrent Accumulate-and-Fire (RAF) neurons' inherent memory to update the cumulative product of keys and values. The segmented attention and lightweight RAF neurons ensure the efficiency of the proposed transformer. Such an approach leads to models with sequential processing capability at a lower computation/memory cost. We apply the proposed method to T5 and BART transformers. The modified models are tested on summarization datasets including CNN-dailymail, XSUM, ArXiv, and Media-SUM. Notably, using segmented inputs of varied sizes, the proposed model achieves $6-22\%$ higher ROUGE1 scores than a segmented transformer and outperforms other recurrent transformer approaches. Furthermore, compared to full attention, the proposed model reduces the computational complexity of cross attention by around $40\%$.

## 1  Introduction

Since the inception of transformers (Vaswani et al., 2017), they have gradually become the de-facto model for Natural Language Processing (NLP) as well as Computer Vision (CV) tasks (Brown et al., 2020; Dosovitskiy et al., 2021). More recently, generative models have also achieved unprecedented results by employing transformers in language and vision for understanding and generation (Saharia et al., 2022; Rombach et al., 2022). One of the key recipes behind the success of transformers is their ability to leverage long-range connections baked in the all-to-all attention mechanism. This sort of attention eases optimization and enables learning of long-term dependency, resulting in improved performance over traditional recurrent neural networks (RNNs). However, transformers incur huge memory and computing requirements, which may not be feasible for resource-constrained applications. Moreover, the computational complexity of transformers increases quadratically with input length, making scaling-up transformers a big challenge. Therefore, improving the efficiency of transformers is a key research direction (Tay et al., 2022). Specifically, for sequence-to-sequence language tasks such as translation and summarization, the decoder part of the transformer inevitably generates words in steps (Guo et al., 2022; Bao et al., 2020). These steps are mostly correlated and hence, some form of recurrence is suited to harness the underlying information efficiently. Without any explicit memory, the vanilla attention processes the same set of previous inputs repeatedly. Such redundant computation can be eliminated by incorporating recurrent units with small memory overhead. As a result, the model can efficiently recapitulate relevant information from past states while computation is performed only on the new inputs.

To that end, there have been efforts to combine transformers with the recurrent mechanism. In TransformerXL (Dai et al., 2019), the long-term dependency is modeled by adding recurrence to transformers. However, this method simply concatenates the hidden states from the last segment with the current state for attention. It does not involve any explicit modification to the attention block itself to better suit the recurrence in order to bridge consecutive segments. Though this is simple to implement, it may not be the best possible way

to merge the attention mechanism with recurrence. Moreover, most existing recurrent transformers are only designed for self attention, whereas, for summarization tasks, cross attention uses much longer inputs and incurs a higher cost. Hence, there is scope to design a custom transformer architecture that can effectively combine recurrence with the embedded attention.

In order to enhance the computational efficiency of transformers, we divide the input sequence into segments to compute *segmented attention*. Next, to aggregate global information over segments, we add *recurrent attention*. We propose recurrent accumulate and fire (RAF) neurons for recurrent attention. It has an implicit memory to accumulate information over segments and only propagates values upon crossing a learnable threshold. By accumulating the partial product of keys and values from other segments, recurrent attention compensates for the loss caused by segmentation. Furthermore, RAF neurons use a leak and a threshold to forget and select information. Compared to RNNs or LSTMs, a RAF neuron has fewer parameters and thus does not add large computational overhead.

Based on the proposal of segmented attention and recurrent attention, we introduce SRformer, as shown in Fig.1. SRformer is a sequence-to-sequence model consisting of an encoder and an auto-regressive decoder. It replaces the decoder's cross-attention with segmented recurrent attention, while self-attention can still attend to all locations of the generated sequence. During training, it selects the corresponding segment for each input and learns recurrent attention between segments. Because recurrent attention processes a segment as a whole, it enables partially parallel computation. During inference, it accesses past information from the memory and only updates the segmented recurrent attention with the current segment. Hence, it is more efficient than full attention. In addition, the number of timesteps for backpropagation depends on the number of segments instead of the length of inputs. The reduced propagation length ensures efficiency and alleviates the vanishing gradient problem. Moreover, we provide a matrix interpretation to demonstrate the harmonious alliance between recurrence and attention obtained through the proposed model.

SRformer achieves better computational complexity and accuracy trade-off. When tested on the CNN-dailymail dataset, the ROUGE1 evalu-

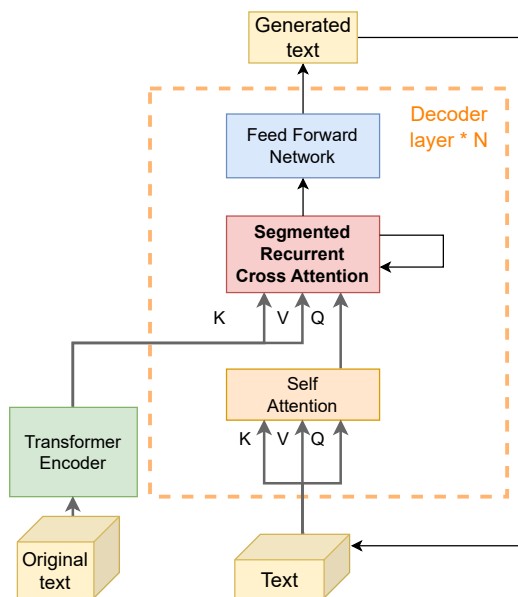

Figure 1: Segmented Recurrent Transformer (SRformer). Cross attention is replaced by segmented recurrent attention.

ation score for the proposed model is $6 - 22\%$ higher than a segmented transformer without recurrent blocks. Interestingly, the performance of SRformer is resilient to extremely small segment sizes (ROUGE1 score drops by only $0.25$ as segment size is reduced from 64 to 8). On the other hand, for the same segment size reduction, the baseline transformer suffers from severe performance degradation (ROUGE1 score drops by $16.04$). Furthermore, the computation reduces by $\sim 40\%$ in comparison with regular cross attention having access to the full sequence. This result clearly demonstrates the efficacy of segmented recurrent attention for sequential data processing, specifically for limited compute budgets.

In the next section, we introduce the related works. In Section 3, we provide details of SRformer. Finally, the experimental results are discussed in Section 4.

## 2   Related Works

Several approaches have been reported in the literature to improve the efficiency of transformers (Tay et al., 2022). Usage of low-rank kernels (Wang et al., 2020), fixed or learnable patterns (Child et al., 2019; Ho et al., 2019; Roy et al., 2021; Kitaev et al., 2020), block-wise processing (Lefaudeux et al., 2022; Dao et al., 2022), recurrent transformers (Dai et al., 2019; Katharopoulos et al., 2020), and global attention (Ainslie et al., 2020; Zaheer

et al., 2020; Beltagy et al., 2020; Guo et al., 2021) have been explored recently.

Our work falls under the category of transformers with recurrence. In this aspect, we closely examine the following two models. TransformerXL (Dai et al., 2019) concatenates the hidden states from the previous segment with the current one. Then each hidden state depends on all previous ones and becomes built-in memory. However, the length of hidden states gets doubled resulting in additional complexity. The Linear Transformer reduces attention to a linear-time, constant memory recurrent neural network (Katharopoulos et al., 2020). It is similar to our recurrent attention without RAF neurons and setting segment size to one. However, linear transformer usually suffers from performance degradation compared to regular attention with softmax. Qin et al. (2022) identified that the devil in linear attention is the scaling of attention matrices and used normalization to stabilize gradients. In our approach, we use RAF neurons and normalization to make a better approximation to softmax attention.

Other types of transformers deal with long sequences by combining local attention with global attention, such as ETC (Ainslie et al., 2020), Big Bird (Zaheer et al., 2020), Longformer (Beltagy et al., 2020), and longT5 (Guo et al., 2021). Each paper proposed a different design for global attention. ETC pretrains global tokens with Contrastive Predictive Coding. Big Bird adds random sparse attention patterns to global attention. Longformer applies global attention on pre-selected locations and makes it symmetric: all locations attend to global tokens and vice versa. LongT5 simply uses the sum of each block as global tokens. Our work substitutes recurrent attention for global attention. We will compare our results with LongT5 since it is the most recent work.

There are two concurrent works about recurrent transformers. The first one is block recurrent transformer (Hutchins et al., 2022). They design recurrent cells to operate on blocks of tokens. Different from our encoder-decoder model, they add cross attention to a decoder-only model, which attends to recurrent states in parallel with self attention. The second work is on recurrent memory transformer (Bulatov et al., 2022). The authors segment inputs similar to our work, then put read and write memory at different ends of the input to attention. However, they only add memory to the input, while

we propose a novel attention mechanism that is a drop-in replacement for regular attention.

Notably, most of these related works focus on the self attention block of the transformer. However, we notice that the cross attention between the encoder and decoder is the bottleneck of computation in summarization tasks. Hence, we focus on improving the efficiency of cross attention for sequence-to-sequence models. Additionally, we use small segment sizes to improve efficiency, while other works use large sizes to improve scalability, which is another distinction between the proposed work and prior art.

## 3 Proposed method

### 3.1 Segmented Attention

Suppose that $Q$ is the query matrix, $K$ is the key matrix, and $V$ is the value matrix. The cross attention weight $A$ and the output attention $O$ are computed by

$$^{q \times k}A = {}^{q \times d}Q *^{d \times k} K^T \qquad (1)$$
$$^{q \times d}O = \text{softmax}(A) *^{k \times d} V \qquad (2)$$

We show the dimension of each matrix at its upper-left superscript. $k$ is the length of encoded features and $q$ is the length of decoder embeddings. $d$ is the inner dimension of each attention head. For summarization tasks, $q$ is usually much smaller than $k$. Note that the batch dimension and the number of heads are ignored for simplicity.

If we split encoded features into segments of size $s$, then the number of segments would be $m = \dfrac{k}{s}$. We define the $i$-th segment of keys and values as $K_{Si} = K[i * s : (i + 1) * s]$ and $V_{Si} = V[i * s : (i + 1) * s]$. To find the corresponding segment for a query at time $t$, we compute the segment index by $i = min(int(t * m/q), m - 1)$.

Let the query at time $t$ be denoted as $Q_t$. Then the full attention weight $A[t]$ and output $O[t]$ at time t are computed using the current query $Q_t$ with full keys and values. However, in the case of segmented cross attention, we use only the corresponding segment of keys and values to compute the attention for the query at time $t$, instead of the full matrix. The segmented attention weight ($A_S[t]$) and output ($O_S[t]$) are computed as in Eqn. 4 and

Eqn. 6.

$$^{1 \times k} A[t] =^{1 \times d} Q_t *^{d \times k} K^T \quad (3)$$

$$^{1 \times s} A_S[t] =^{1 \times d} Q_t *^{d \times s} K_{Si}^T \quad (4)$$

$$^{1 \times d} O[t] = \text{softmax}(A[t]) *^{k \times d} V \quad (5)$$

$$^{1 \times d} O_S[t] = \text{softmax}(A_S[t]) *^{s \times d} V_{Si} \quad (6)$$

Unlike directly projecting $K$ to a lower rank of size $d \times s$ for all timesteps, each query is multiplied with a different segment $K_{Si}$ and $V_{Si}$. Hence every segment is used at least once during the process and there are no extra weights or information loss caused by projection. The computation complexity is reduced to $O(qds)$ from $O(qdk)$, where $s << k$. If $s$ is equal to the summarization ratio $k/q$, then the computation complexity is reduced to $O(kd)$ from $O(k^2 d/s)$ and no longer grows quadratically with the length. Segmented attention described above largely improves efficiency but at the loss of the ability to access all information, which causes accuracy degradation. Therefore, next, we focus on how to compensate for the loss and achieve a better trade-off between efficiency and accuracy using recurrent units between segments.

### 3.2 Approximating full attention with segmented recurrent attention

The segmented attention described in the last subsection simply replaces $K$ and $V$ with the current segments $K_{Si}$ and $V_{Si}$, but it neglects a large part of the keys and values. Therefore, we need to compensate for the loss to better approximate the full attention.

Suppose we divide the inputs into two parts: the current segment $S_i$ and the remaining segments $R_i$. By linearity of matrix operations, if we switch the order of columns in $K$ and that of corresponding rows in $V$, it does not affect the matrix multiplication result. Hence, we can partition matrix $K$ into $\begin{bmatrix} K_{Si} & K_{Ri} \end{bmatrix}$ and $V$ into $\begin{bmatrix} V_{Si} & V_{Ri} \end{bmatrix}^T$, where $K_{Ri}$ and $V_{Ri}$ are the remaining parts of the matrices excluding the current segment. The output $O$ of full attention is computed by

$$A = Q * K^T = Q * \begin{bmatrix} K_{Si} & K_{Ri} \end{bmatrix} \quad (7)$$

$$O = \text{softmax}(Q * \begin{bmatrix} K_{Si} & K_{Ri} \end{bmatrix}) \begin{bmatrix} V_{Si} \\ V_{Ri} \end{bmatrix} \quad (8)$$

By expanding Eqn. 8, full attention at time $t$ can be rewritten as the weighted sum of segmented attention $O_S[t]$ and the attention of remaining parts

$$O_R[t] = \text{softmax}(Q_t * K_{Ri}^T) V_{Ri}.$$

$$O[t] = \frac{c_s}{c} O_S[t] + \frac{c - c_s}{c} O_R[t] \quad (9)$$

$$O[t] = \sigma O_S[t] + (1 - \sigma) O_R[t] \quad (10)$$

where $c$ and $c_s$ are divisors of softmax. They can be replaced by a weighing factor $\sigma$. Please see the appendix for detailed analyses.

In the proposed approach, we approximate the output at time $t$ by computing the segmented attention $O_S$ of the local segment using Eqn. 6, then adding the approximated $O_R$ for other segments. The weighing factor $\sigma$ can be a trainable parameter. Note, the simplified sum shown in Eqn. 11 still works because of the automatic adjustment of other parameters.

$$O[t] = O_S[t] + O_R[t] \quad (11)$$

To efficiently approximate the compensation part $O_R$, we propose *recurrent attention*. This is important because if we directly compute it, the total computational complexity becomes the same as a non-segmented model. Inspired by linear attention proposed in Katharopoulos et al. (2020), we multiply keys and values first, then multiply with queries. This is validated by the associativity property of matrix multiplication. The switch of order allows storing the product of keys and values as a state of RNN. The softmax operation applied on the attention weights is approximated as-

$$O_R[t] = \text{softmax}(Q_t * K_{Ri}^T) V_{Ri} \quad (12)$$

$$\approx \frac{Q_t * RAF(K_{Ri} * V_{Ri})}{norm(K)} \quad (13)$$

The $RAF$ in this equation is a recurrent unit designed to accumulate sequential information. It is explained in detail in subsection 3.3. In Eqn. 13, division by $norm(K)$ is used to simulate the normalization factor of softmax.

We simplify the computation of multiplying $K_{Ri}$ and $V_{Ri}$ by the multiplication rule of the matrix, as follows. The product is denoted by $P$.

$$K * V = \begin{bmatrix} K_{Si} & K_{Ri} \end{bmatrix} \begin{bmatrix} V_{Si} \\ V_{Ri} \end{bmatrix} \quad (14)$$

$$\Rightarrow K * V = K_{Si} * V_{Si} + K_{Ri} * V_{Ri} \quad (15)$$

$$P_i = K_{Ri} * V_{Ri} = K * V - K_{Si} * V_{Si} \quad (16)$$

Thus, we can easily reduce the computational complexity of Eqn. 16 from $O(d * (k - s) * d)$ to $O(d * s * d)$. For $P_i$ of different segments, we only need to compute $K * V$ once. Fig. 2 shows the computation process of recurrent cross attention.

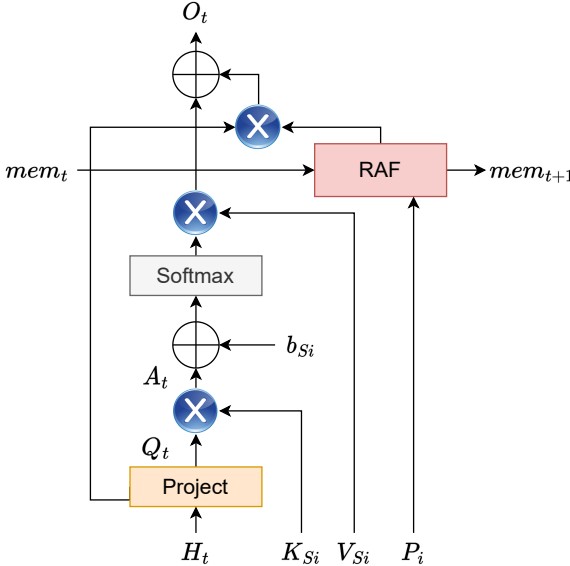

Figure 2: Segmented Recurrent Attention in a cross attention block. $K_{Si}$ and $V_{Si}$ are segmented keys and values. $H_t$ is the hidden input, which is projected to $Q_t$ by a linear layer. $b_{Si}$ is the bias of T5 model.

## 3.3 Recurrent Accumulation-and-Fire neuron

RAF neurons accumulate the partial products of keys and values, help approximate softmax, and filter out unnecessary information. It is inspired by Integrate-and-Fire neurons from biological brains and Spiking Neural Networks (Lee et al., 2016; Wozniak et al., 2020; Ponghiran and Roy, 2021; Chowdhury et al., 2022). The architecture of an RAF neuron is shown in Fig. 3. We also provide Pytorch-like pseudo code for RAF in Appendix A.2.

Each RAF neuron uses a membrane potential (memory) to efficiently aggregate sequential information. At each timestep, the memory is updated by adding the projected input with the previous memory, where memory gets decayed by a learnable leak. The input to RAF neurons is the partial product $K_{Ri} * V_{Ri}$ containing information of the sequence except the current segment. It is projected by a linear layer. Since Eqn. 13 does not use softmax, the linear layer is necessary to adjust the scale of inputs. Next, we check if its memory crosses the threshold of information propagation to the next layer. If yes, the information in its memory gets passed to the next processing layer, else, the output is zero. This is activated by applying ReLU on the thresholded memory. The learned threshold works like a bias to activate only values over it. If the memory has activated firing, it will be

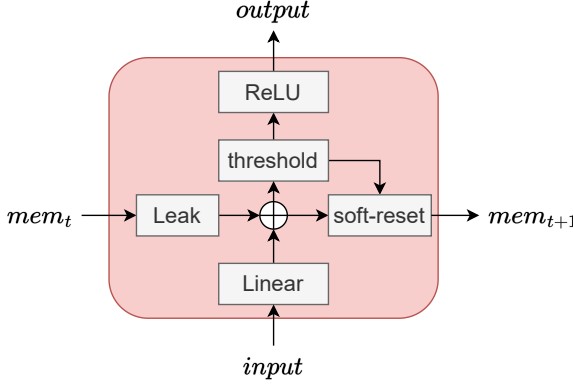

Figure 3: Recurrent Accumulation-and-Fire neuron. The leaky factor and threshold are trained parameters.

soft reset by subtracting the value of the threshold. Then it will not fire again until new information is given. Because $K_{Ri}$ and $V_{Ri}$ at different steps have overlaps, reset is important to forget redundant information. The design of RAF results in a sparse and low-complexity bio-inspired recurrence mechanism. Unlike an LSTM, a RAF neuron does not apply weights on the hidden memory, but only uses a learnable leaky factor, a threshold, and soft reset as forget and output gates. Hence, it requires fewer weights and lesser computation compared to an LSTM.

---

**Algorithm 1:** One step of cross attention blocks in SRformer

---

**Input:** Query $Q_t$; Keys $K$ and Values $V$ projected from encoded features; Timestep $t$; previous index $prev$;

**Output:** Output $O_t$

$K_s$ = split($K$); $V_s$ = split($V$)

$i = min(int(t * m/q), m - 1)$

$A[t]$ = matmul($Q_t, K_s[i]$)

$A[t]$ = softmax($A[t]$)

**if** $t == 0$ **then**
  $\quad$ $KV$ = matmul($K$,$V$)
  $\quad$ memory = zero tensor
**end**

$O_S$ = matmul($A[t], V_s[i]$)

**if** $prev != i$ **then**
  $\quad$ $P$ = $KV$ - matmul($K_s[i]$,$V_s[i]$)
  $\quad$ $P$, mem = RAF($P$, mem)
**end**

$O_R$ = matmul($Q_t, P$)/norm($K$)

$O_t = O_S + O_R$

$prev = i$

---

| | Comp. complexity | Example (K) | Memory complexity | Example (K) |
|---|---|---|---|---|
| **Transformer** | $O(qkd)$ | $qkd = 8389$ | $O(qk + kd)$ | 197 |
| **Segmented** | $O(qsd)$ | $qsd = 524$ | $O(qs + kd)$ | 74 |
| **SRformer** | $O(qsd + kd^2)$ | $qsd + kd^2 = 4718$ | $O(qs + kd + md^2)$ | 139 |

Table 1: Computation and memory complexity of cross attention. For example, in one of our experiments on the CNN-dailymail dataset, $q = 128$; $k = 1024$; $d = 64$; $s = 64$. To illustrate theoretical computation and memory cost, we show the product of example sizes in thousands.

### 3.4 Segmented Recurrent Transformer

SRformer aims at utilizing the sequential generation process to capture the relation between locations of a long sequence, without paying attention to all locations or propagating from start to end. After reducing the length of keys and values to reduce computational complexity, it uses Eqn. 13 to approximate the attention mechanism.

Algorithm 1 shows the process of one step of cross-attention blocks in SRformer. First, keys and values are split into segments. Then it computes the attention score $A[t]$ for the current query with the corresponding segment of keys $K_{Si}$. Because the length of queries is usually greater than the number of segments, some queries might use the same segment. Only when $t = 0$, which means it is the beginning of a new sequence, it initializes the memory of RAF and compute the full product of $KV$. After applying softmax, attention scores are multiplied with segmented values to get the segmented attention output $O_S$. It is important to compute the partial product $P$ of keys and values only once for each index. In addition, the computation of $P$ can be parallelized to improve efficiency. Then $P$ is given to the RAF layer for accumulating with the memory through the recurrent process. The output of RAF is multiplied by the query and then normalized. Finally, we add the segmented attention output $O_S$ and the complementary $O_R$.

Our approach builds connections between segments by using the partial product of keys and values as memory. Since the RAF neuron is applied between segments, the backpropagation-through-time in our model is more efficient than using recurrent layers on each input. As discussed before, recurrent attention makes up for the loss caused by segmented attention. Together, they approximate a complete attention mechanism at a lower cost. Although some existing works proposed global attention for a similar purpose, global attention based on selected locations cannot make up for the loss.

Since segmented recurrent attention is a drop-in replacement for attention, it can be used in all encoder-decoder transformers, such as T5 and BART. Please note that the usage of positional bias and masks varies across transformers and needs modification. We provide details in Appendix. A.3.

### 3.5 Complexity Analysis of cross attention

The total complexity of a sequence-to-sequence transformer is composed of the cost of the encoder and that of the decoder. Here, we only compare the complexity of cross-attention since the other components are not affected by the proposed method. In table. 1, we show both computation and memory (space) complexity analyses of a regular transformer, a segmented one, and the proposed SRformer. Please note that the batch size and the number of heads are not considered. To compute the segmented qkv attention, the complexity is $O(qsd)$. The bottleneck for computation is to multiply $K_{Si} * V_{Si}$ for $m$ times, which results in $O(msd^2)$ complexity. Because the number of segments is $m = k/s$, it becomes $O(kd^2)$. After that, we need to multiply $Q$ with it, which costs $O(qd^2)$, but as $q << k$, this term can be ignored. The recurrent unit also costs much less computation than the bottleneck, so it can also be ignored. Since the segment size $s$ and the inner dimension $d$ of each head are much smaller than $k$ in summarization datasets, the number of computations is fewer than a regular transformer, as shown in table 1.

The memory complexity depends on the sizes of input matrices and intermediate matrices. The size of key and value matrices are $k \times d$. The intermediate matrix is the attention weight $A$, whose size is $q \times k$. By segmentation, its size is reduced to $q \times s$. For recurrent attention, we compute the product of keys and values, $P$, for $m$ segments. Hence, there are $m$ matrices of size $d \times d$. In table. 1, we show that given the example sizes, the theoretical reduction of computation is 43%, and that of memory complexity is 29%.

## 4 Experimental Results

We test our proposed model and method on T5 sequence-to-sequence transformers (Raffel et al., 2022) and BART model (Lewis et al., 2020). Since we want to design a model for resource-constrained environments, we use T5-small, which contains 6 transformer layers and 60 million parameters, and BART-base which contains 6 layers and 140 million parameters. Our implementation is built based on Huggingface transformer platform, which is released under the Apache-2.0 License, and we load the pretrained models before training (Wolf et al., 2019). The implementation is based on PyTorch python package (Paszke et al., 2019).

We run all experiments on Nvidia A40 GPUs. Each experiment is run on a single core to compare GPU power and memory usage. We train the models for 25 epochs as the loss and evaluation scores converge at that point. The initial learning rate is $5e^{-5}$. The optimizer is AdamW and the learning rate scheduler is linear with default settings.

We use ROUGE scores to evaluate SRformer and compare it with other transformers. ROUGE-N is defined as the n-gram recall between model predictions and a set of ground-truth summaries, and ROUGE-L measures the longest common subsequence between them (Lin, 2004). Note, a higher ROUGE score denotes better summarization performance.

### 4.1 Results on summarization datasets

The summarization datasets we use to evaluate the model include CNN-dailymail (Nallapati et al., 2016), XSUM (Narayan et al., 2018), ArXiv Summarization (Cohan et al., 2018), and MediaSum (Zhu et al., 2021). CNN-dailymail dataset contains 311k anonymized English news articles and their summaries, among which 286,817 are training pairs, 13,368 are validation pairs, and 11,487 are test pairs (Hermann et al., 2015; See et al., 2017). XSUM is an extreme summarization dataset, and thus its summaries only have 23 words on average (Narayan et al., 2018). The numbers of documents for training, validation, and test sets of XSUM dataset are 204,045/11,332/11,334. ArXiv dataset has 215 thousand documents, including 5% for validation and another 5% for test (Cohan et al., 2018). MediaSUM is a media interview dataset consisting of 463.6K dialogue transcripts and summaries, and validation and test set each randomly select 10K instances (Zhu et al., 2021).

Table 2 lists the detailed results including ROUGE1, ROUGE2, ROUGEL, and ROUGEL-Sum scores. From the table, we can see that a model with a higher ROUGE1 score usually gives higher other scores, and hence can be considered as better. The sizes of keys and values for cross attention are 1024. With the default segment size set to 64, the sizes of keys and values are reduced by 16. As a consequence, both T5 and BART experience large performance degradation. Then we show ROUGE1 scores of SRformer in bold numbers. SRformer benefits from its recurrent attention mechanism and achieves $3 - 10\%$ higher scores than the segmented transformers. Although its ROUGE scores are slightly lower than the baseline model, the computational complexity is largely reduced, and we will show it also cuts down GPU power usage and memory access time in section 4.4.2.

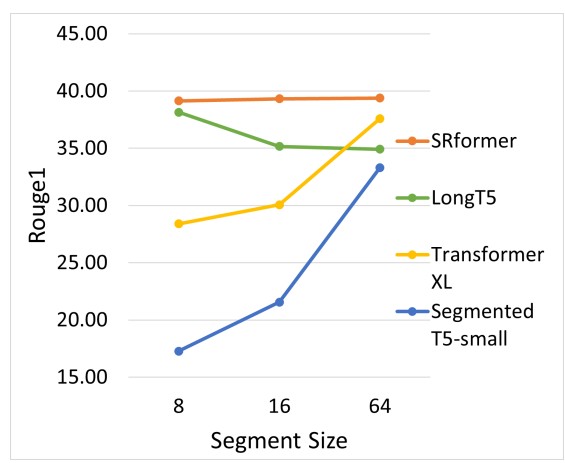

Figure 4: Rouge1 scores on CNN-dailymail dataset for different segment sizes across various models.

### 4.2 Effect of segment sizes

Figure 4 shows ROUGE1 scores on CNN-dailymail dataset with regard to different segment sizes 8, 16, and 64. The number of segments should not exceed the number of generation steps, otherwise, some segments would never be used. Hence, the segment size should not be smaller than 8 for the CNN-dailymail dataset.

The blue line in Fig. 4 shows that for a segmented transformer without recurrence, the ROUGE1 score drops significantly by 16% when reducing segment size from 64 to 8. The yellow line shows the results of applying the method from TransformerXL (Dai et al., 2019) to cross attention blocks of the T5-small model. We add a hidden memory for keys and values and concatenate

| Dataset | Model | SRformer | | | | Segmented | Baseline |
| | | ROUGE2 | ROUGEL | ROUGELsum | ROUGE1 | ROUGE1 | ROUGE1 |
|---|---|---|---|---|---|---|---|
| CNN-DM | T5 | 16.79 | 27.29 | 36.81 | **39.39** | 33.31 | 41.60 |
| | BART | 19.80 | 29.44 | 40.40 | **43.19** | 33.87 | 44.54 |
| XSUM | T5 | 11.72 | 26.88 | 26.88 | **34.48** | 27.03 | 35.35 |
| | BART | 16.58 | 31.54 | 31.54 | **39.03** | 36.22 | 41.33 |
| ArXiv | T5 | 11.50 | 24.53 | 31.86 | **36.04** | 32.64 | 37.79 |
| | BART | 14.97 | 25.29 | 37.86 | **42.99** | 36.43 | 43.95 |
| MediaSum | T5 | 12.89 | 25.72 | 26.29 | **28.59** | 24.43 | 29.25 |
| | BART | 14.70 | 24.75 | 28.46 | **32.36** | 28.40 | 33.88 |

Table 2: Results of SRformer on Summarization datasets compared to baseline models and their segmented version. The baseline models are T5-small and BART-base.

the hidden memory with the current segment to compute cross attention. With that setting, TransfomerXL performs better than a segmented transformer, but its performance still decreases with the segment size. The green line shows the results of longT5 (Guo et al., 2021). When the number of segments grows as the segment size decreases, it can use more tokens for computing global attention. Hence, its performance is inversely proportional to the segment size. For example, when segment size is 8, it uses 8 local and 128 global tokens, and its performance is 3% better than using 64 local and 16 global ones. However, the computational overhead of global attention also grows quadratically with the number of tokens. In contrast, SRformer is able to keep a stable performance regardless of the size reduction. Its ROUGE1 scores are $6 - 22\%$ higher than those of a segmented T5 model, and it gives a larger boost in performance compared to TransformerXL and longT5. Even when other models use a segment 8 times larger, they are still not able to compete with our model. The results show that SRformer can indeed better approximate full attention.

| Setting | ROUGE1 | Setting | ROUGE1 |
|---|---|---|---|
| SRformer | 39.39 | - | - |
| (I) | 36.86 | (II) | 36.77 |
| (III) | 35.84 | (IV) | 29.72 |

Table 3: Ablation study using CNN-Dailymail dataset

## 4.3 Ablation Study

To better understand the contribution of each component to the performance of SRformer, we conducted an ablation study by removing or modifying terms in Eqn. 11 or 13. The segment size is fixed to

64 and the model is modified from T5-small. The dataset used for ablation study is CNN-dailymail. We varied the following components one by one while keeping the rest of the model fixed:

(I) Remove segmented attention. We modify the model to only use recurrent attention. We observe a noticeable drop in performance, however, it is much better than only using segmented attention. This shows that recurrent attention made a crucial contribution to the overall performance.

(II) Remove normalization. Without dividing $norm(K)$, ROUGE 1 score drops by 3.

(III) Without RAF. Skipping RAF will remove recurrent computation and corresponding weights. However, without RAF, it is not able to approximate full attention well. The performance decrease shows the importance of RAF.

(IV) Replacing $K_R * V_R$ with $K_s * V_s$. In this case, it can skip computing $K * V$. RAF accumulates information as timesteps increase. However, at the beginning of generation, it only has information from the first few segments. Hence the performance is not as good.

In summary, our ablation study shows that all terms in Eqn. 13 are important components.

## 4.4 Evaluation of improvement in efficiency

### 4.4.1 Theoretical improvement

In Table.1, we have shown that SRformer reduces the computation and memory complexity of cross attention. In our experiment, as the length of the generated summary $q$ equals 128 and the length of the input article $k$ equals 1024, the theoretical reduction of computation is 43%, and that of memory complexity is 29%. In Fig. 5, we show how the improvement would change when the length of the generated summary scales up. The costs are

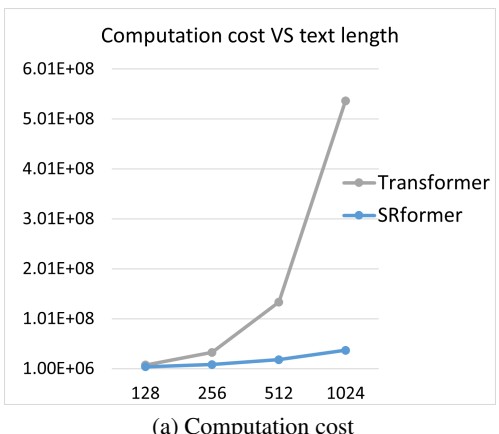

(a) Computation cost

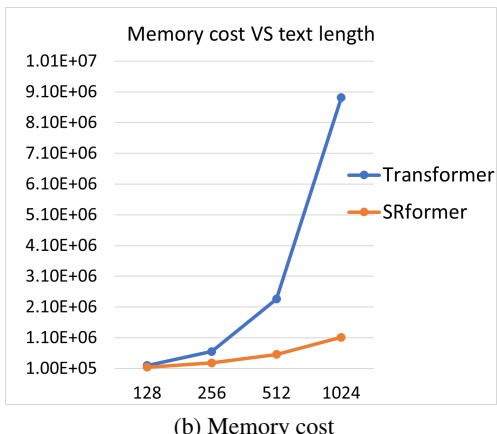

(b) Memory cost

Figure 5: Theoretical computational cost and memory cost of cross attention when the length of summary ($q$) increases from 128 to 1024.

computed based on the theoretical complexity in Table.1. Suppose that the length of the input article is eight times the length of the summary ($k = 8 * q$). The cost of cross attention in a regular transformer will increase quadratically with the text length. The cost of cross attention in SRformer is not dependent on $q * k$ but $q * s$, where $s$ is a small segment size. Therefore, it increases linearly and the reduction would be 93% compared to a transformer when $q = 1024$. Although existing datasets for abstractive summarization have relatively short summary lengths, large language models may be required to generate a long summary for a very long document at a fixed ratio. In that case, SRformer can provide a large improvement in efficiency.

### 4.4.2 GPU power and memory usage

We track GPU memory, memory accessing time, and power using Weights&Biases tools (Biewald, 2020). The GPU usage of SRformer modified from BART and other models are listed in Table. 4. Our model consumes less GPU power and spends less time accessing memory, compared to a regular transformer, TransformerXL (Dai et al., 2019), and LongT5 (Guo et al., 2021). BART uses 99% GPU power, which almost reaches the limitation of GPU. The percentage of time GPU spent accessing memory is 91%. It means GPU spent most of the time fetching data from memory. However, with segmented recurrent attention, SRformer does not need to use all encoded features at each step. Hence it saves a considerable amount of computation and memory access. It only uses 81% GPU power and 60% GPU memory accessing time. Although GPU memory usage of all recurrent transformers

| Model | Mem | Time | Power |
|---|---|---|---|
| BART | 49 | 91 | 99 |
| TransformerXL | 82 | 70 | 89 |
| LongT5 | 73 | 80 | 90 |
| SRformer | 60 | 60 | 80 |

Table 4: Percentages (%) of GPU usage in terms of memory, memory accessing time, and power usage

is larger than vanilla BART to store weights and gradients, our method uses less memory than TransformerXL and LongT5. In conclusion, by alleviating the cross attention bottleneck, the proposed transformer becomes more efficient.

## 5 Conclusion

In this paper, we propose an efficient sequence-to-sequence transformer using segmented recurrent attention. Since cross-attention is the computation bottleneck for text summarization, replacing full attention with segmented recurrent attention can largely reduce its computation complexity. The proposed RAF neurons help approximate full attention at a low overhead. During inference, they utilize the sequential process to accumulate information in their internal memory and bridge information across segments effectively. The efficacy of the proposed model is demonstrated through higher ROUGE scores on multiple summarization datasets compared to other segmented transformers. Our proposed architecture is especially suitable for summarization and can be used for other sequence-to-sequence tasks such as translation and question-answering.

## Limitations

This work focuses on designing an efficient encoder-decoder model for summarization tasks. The proposed method can not be directly applied to an encoder-only or decoder-only model, although it is possible to design a modified version for those models. In the case that the length of encoded features is not much larger than the length of decoder output, it may not be able to show significant reduction in computation and memory cost.

## Ethics Statement

This work complies with the ACL Ethics Policy.[1] It contributes a new model to the deep learning and computational linguistics community which can be used in many applications. We will make the codes open-source for other researchers.

## Acknowledgement

This work was supported by the Center for Codesign of Cognitive Systems (CoCoSys), one of the seven centers in JUMP 2.0, a Semiconductor Research Corporation (SRC) program sponsored by DARPA.

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

# A    Appendix

## A.1    Expanding attention equations

Denote the current i-th segment as $S_i = [i * s, (i + 1)s]$. The indices of remaining segments are denoted as $R_i = \{j \in [1, k], j \notin S_i\}$. Then the full attention, segmented attention, and remaining attention at timestep $t$ will be

$$O[t] = \text{softmax}(Q_t * K^T)V$$
$$O_S[t] = \text{softmax}(Q_t * K_{Si}^T)V_{Si}$$
$$O_R[t] = \text{softmax}(Q_t * K_{Ri}^T)V_{Ri}$$

For example, the softmax operation applied on the remaining attention $O_R$ can be expanded as

$$O_R = \text{softmax}(Q_t * K_{Ri}^T)V_{Ri} \qquad (17)$$

$$= \frac{\phi(Q_t * K_{Ri}^T) * V_{Ri}}{\sum_{j \in R_i} \phi(Q_t * K_j^T)} \qquad (18)$$

$$= \frac{\sum_{j \in R_i} (\phi(Q_t * K_j^T)V_j)}{\sum_{j \in R_i} \phi(Q_t * K_j^T)} \qquad (19)$$

In the common implementation of softmax attention, $\phi(X) = \exp(X)$ is an exponential function. For the numerical stability of the computation, another choice is $\phi(X) = exp(X - log(\max_i X_i))$ (Li, 2022). However, Katharopoulos et al. (2020) proposed to replace it with a kernel function such that $\phi(XY) = \phi(X)\phi(Y)$. Then we have

$$O[t] = \frac{\phi(Q_t) \sum_{j=1}^k (\phi(K_j^T)V_j)}{\sum_{j=1}^k \phi(Q_t * K_j^T)} \qquad (20)$$

$$O_S[t] = \frac{\phi(Q_t) \sum_{j \in S_i} (\phi(K_j^T)V_j)}{\sum_{j \in S_i} \phi(Q_t * K_j^T)} \qquad (21)$$

$$O_R[t] = \frac{\phi(Q_t) \sum_{j \in R_i} (\phi(K_j^T)V_j)}{\sum_{j \in R_i} \phi(Q_t * K_j^T)} \qquad (22)$$

Denote the divisor in the above equations using $c_s, c_r$, and $c$.

$$c_s = \sum_{j \in S_i} \phi(Q_t * K_j^T)$$

$$c_r = \sum_{j \in R_i} \phi(Q_t * K_j^T)$$

$$c = \sum_{j=1}^k \phi(Q_t * K_j^T) = c_s + c_r$$

Then we can write the full attention as a weighted summation, where the weights depend on $c$ and $c_s$.

We can use a factor $\sigma$ to represent the weights.

$$O[t] = \frac{c_s}{c}O_S[t] + \frac{c_r}{c}O_R[t]$$

$$O[t] = \sigma O_S[t] + (1 - \sigma)O_R[t]$$

In practice, we find that simple addition without weighting still works. Our recurrent attention further approximates $O_R$ as below.

$$O_R[t] \approx \frac{Q_t * RAF(K_R^T * V_R)}{norm(K)}$$

Moveover, we can estimate the error caused by segmented attention.

$$O - O_S$$

$$= \frac{c_s * \phi(Q_t * K^T)V - c * \phi(Q_t * K_{Si}^T)V}{c * c_s}$$

$$= \frac{c * \phi(Q_t * K_R^T)V_R - c_r * \phi(Q_t * K^T)V}{c * (c - c_r)}$$

$$= \frac{c_r}{c - c_r}(O_R - O) = \frac{c_r}{c}(O_R - O_S)$$

## A.2 Pseudo code for RAF neurons

```
class RAF(nn.Module):
  def init(dim):
    self.linear = Linear(dim, dim)
    self.activation = ReLU
    self.leak = parameter(1.0,
        require_grad=True)
    self.thre = parameter(0.1,
        require_grad=True)

  def forward(input, mem, t):
    x = self.linear(input)
    mem = leak * mem + x
    y = mem/self.thre - 1.0
    mem = mem - self.thre*(y>0)
    out = ReLU(y)
    return out
```

## A.3 Positional bias and masks

Different transformers such as T5 and BART use positional bias and masks differently. Hence, we need to take this into consideration when implementing segmented recurrent attention.

For example, T5 adds positional bias to the attention weights $A$. For segmented attention, we need to segment bias as shown in the following equations.

$$b_{Si} = b[i * s : (i + 1) * s]$$

$$O_S[t] = softmax(A_S[t] + b_{Si}) * V_{Si}$$

BART adds positional bias to the inputs of the first layer, so there is no need to take care of that separately. However, the encoder of BART generates features of various sizes and adds padding to each batch. To ignore the padding, an encoder mask is used. Because softmax is removed in recurrent attention, adding negative infinity to masked locations is not suitable. Instead, we filter out abnormal values of $K * V$ and $P$ with a clamp function before computing recurrent attention.

To apply SRformer to other types of transformers, we recommend checking the positional bias and masks and adjusting accordingly.