# OpenReview forum: "Segmented Recurrent Transformer: An Efficient Sequence-to-Sequence Model"
_EMNLP/2023/Conference — EMNLP 2023 Findings_

### Official Review · Reviewer_qsxJ · 2023-07-29

**Typos Grammar Style And Presentation Improvements:** The format of the rouges in Table 2 d…
**Soundness:** 4

**Excitement:**

3: Ambivalent: It has merits (e.g., it reports state-of-the-art results, the idea is nice), but there are key weaknesses (e.g., it describes incremental work), and it can significantly benefit from another round of revision. However, I won't object to accepting it if my co-reviewers champion it.

**Paper Topic And Main Contributions:**

Transformers have exhibited exceptional performance in various domains, including language and vision. However, their computational cost increases quadratically with sequence length, which makes them unsuitable for resource-constrained applications. To address this issue, the paper proposes an alternative approach by dividing the entire sequence into segments and employing local attention mechanisms on each segment.

Segmented Recurrent Transformer (SRformer), combines segmented (local) attention with recurrent attention. By using recurrent attention, the loss incurred by reducing the attention window length is compensated and information across segments is aggregated effectively. The SRformer utilizes Recurrent Accumulate-and-Fire (RAF) neurons, which possess inherent memory, to update the cumulative product of keys and values. The combination of segmented attention and lightweight RAF neurons ensures the efficiency of the proposed transformer.

**Questions For The Authors:**

> line 358 "Unlike RNNs, RAF does not apply weights on the hidden memory but only uses a learnable leak and threshold. Hence, it requires fewer weights and lesser computation compared to an RNN or LSTM."

Does following the LSTM style not improve performance?

**Reasons To Accept:**


SRformer achieves a superior trade-off between computational complexity and accuracy. It reduces computation by approximately 40% when compared to regular cross-attention models that have access to the full sequence. This outcome demonstrates the effectiveness of segmented recurrent attention for processing sequential data, particularly when dealing with extremely limited computational resources.

**Reasons To Reject:**

Although RAF neurons are introduced as a key element of SRformer, the paper lacks an in-depth analysis of their working mechanisms and limitations. Providing more insights into how RAF neurons affect model performance would improve the overall contribution of the paper.

**Reproducibility:**

4: Could mostly reproduce the results, but there may be some variation because of sample variance or minor variations in their interpretation of the protocol or method.

**Reviewer Confidence:**

3: Pretty sure, but there's a chance I missed something. Although I have a good feel for this area in general, I did not carefully check the paper's details, e.g., the math, experimental design, or novelty.

---

> ### Author Rebuttal · Authors · 2023-08-29
>
> We appreciate the reviewer’s positive feedback.
>
> **RAF Neuron**: In SRformer, we propose Recurrent-accumulate-and-fire (RAF) neurons for implementing recurrent attention. The temporal dynamics of a RAF neuron can be expressed as
>
>     U[t]=λ*U[t-1]+linear(X[t]) (i),
>     Y[t]=U[t]/V_th -1.0 (ii)
>     if U[t]>V_th,O[t]=Y[t],U[t]=U[t]-V_th (iiiA)
>     else,O[t]=0, (iiiB)
>
> where U is the accumulated membrane potential, λ is the learnable leak, V_th is the learnable threshold parameter, X is the input, Y is just a scaled and shifter version of U and O is the output. This RAF neuron is conceptually inspired by Leaky-Integrate-and-Fire(LIF) neurons of biological brains [1][2], which can efficiently aggregate sequential information. At each timestep, the membrane potential U[t] is updated by adding the incoming input X[t] with the previous information U[t-1], where U[t-1] gets decayed by a learnable leak λ. Next, we check if the accumulated U[t] crosses the threshold (V_th) of information propagation to the next layer. If yes, that accumulated information Y[t] gets passed to the next processing layer (and V_t is subtracted from the U[t] as a reset of potential), else the output is 0 (and U[t] is retained). This results in a sparse and low-complexity bio-inspired recurrence mechanism. Note, the difference between LIF and RAF is that LIF integrate binary input spikes and propagates discrete output spikes, whereas RAF accumulates the partial products of keys and values and uses ReLU activation. The pseudo-code for RAF neurons is provided in the submitted manuscript (appendix section A.2) and the schematic of their working mechanism is given in Fig. 3.
>
> A limitation of RAF neurons is that modeling very long sequences and very complicated dependencies can be tricky using them. However, as our results demonstrate, they can provide highly efficient solutions as a low-cost recurrent block for processing segments and achieve results comparable to the costly baselines on popular summarization benchmarks.
>
> **RAF versus LSTM/RNN**: We have tried replacing RAF with RNN and the results show that RNN does not significantly improve the performance (ROUGE1 ±0.2), although it uses extra weights on hidden states. Hence, we believe the recurrent attention is important for making up the loss caused by segmentation, and a simple recurrent unit can do the work. We have not tried LSTM because it takes four times more parameters/computation than RNN. Similar results are reported in [2], where LSTMs provide only marginal performance improvements compared to accumulate-and-fire type activations, however the parameter count becomes 2X, with increased compute. As a result, the RAF can be regarded as a lightweight and efficient recurrent unit in lieu of LSTM cells, where the leak ($\lambda$) of RAF acts as a simple forget mechanism and the thresholding operation ($V_{th}$) leads to sparse and efficient information propagation.
>
> The effect of the RAF neurons is elucidated in Table 3, case (III) of the submitted paper, where the ROUGE1 score drops by 4.0 when RAF is not used. Furthermore, we have tested how the design of RAF would affect the model performance. For example, we have tried to remove the linear layer, replace ReLU with binary spike activation used in LIF, or replace the linear layer with element-wise multiplication with learnable parameters, they all perform worse than the current design of RAF (ROUGE1 drop to 32.62%, 32.63%, and 28.52%, respectively, on CNN-dailymail). Due to the page limitation, we only showed the final design of RAF that gives the best results. We will add these ablation studies and an in-depth analysis of RAF in the appendix to the camera-ready version.
>
> For the typo regarding the format of the rouges in Table 2, we will capitalize ROUGE scores to be uniform.
>
> **Additional comparison tables**
>
> Regarding the computational reduction, we improve the computational complexity from O(qkd) to O(qsd + kd^2). In many cases, the length of summary, q, depends on the length of article, k. Then the improvement is from quadratic to linear. The computational reduction increases proportionally with the lengths, as shown in Table A, which is an extension for Table 1 in our paper. When q=128 and k=1024, the overall cost is reduced by 43.8%. When they both increase by 4 times, the improvement will be 85.9%. Moreover, most existing works only reported the computation reduction of attention and ignored the other layers such as feed forward layers in overall cost.
>
> We have recorded the actual memory footprints and GPU usage of training different models as shown in Table B. Our method uses the lowest percentages of GPU power usage and memory accessing time, providing 29%/31% improvement over the baseline. Although GPU memory usage of all recurrent transformers is larger than vanilla BART to store weights and gradients, our method uses less memory than TransformerXL and LongT5. Considering these, we believe our improvement is significant. Besides, we have provided experiments on four different datasets in Table 2 in our paper and an ablation study in section 4.3 to validate the effectiveness of the method.
>
> **Table A: Number of computation of cross attention VS length**
> |     length	q/k	    |    128/1024	|        256/2048	   |     512/4096|
> |---------------|-------------------|--------------------|--------------|
> |Transformer (full attention)|	8.39E+06 |	3.36E+07 |	1.34E+08|
> |SRformer	    |    4.72E+06 |	9.44E+06 |	1.89E+07|
> |reduction	     |   43.8%	|	71.9%	|	85.9%|
>
> **Table B: Memory and power comparisons**
> |			   |     GPU Mem(%)	| GPU mem access time(%)	|Power usage(%)	 |
> |---|---|---|---|
> |BART	(Baseline)	|	49		|	91			|	99|
> |TransformerXL		|	82		|	70			|	89|
> |LongT5				 |       73		|	80			|	90|
> |SRformer - BART		  |      60		|	60			|	80|
> |SRformer - T5		|	        62	|		60		|		81|
>
>
> [1] Lee, J. H., Delbruck, T., & Pfeiffer, M. (2016). Training deep spiking neural networks using backpropagation. Frontiers in neuroscience, 10, 508.
>
>  [2] Ponghiran, W., & Roy, K. (2022, June). Spiking neural networks with improved inherent recurrence dynamics for sequential learning. In Proceedings of the AAAI Conference on Artificial Intelligence (Vol. 36, No. 7, pp. 8001-8008).

---

### Official Review · Reviewer_yFAe · 2023-07-31

**Soundness:** 3

**Excitement:**

3: Ambivalent: It has merits (e.g., it reports state-of-the-art results, the idea is nice), but there are key weaknesses (e.g., it describes incremental work), and it can significantly benefit from another round of revision. However, I won't object to accepting it if my co-reviewers champion it.

**Paper Topic And Main Contributions:**

This paper proposed a segmented recurrent transformer that combines segmented attention with recurrent attention. This method achieved higher scores than other approaches and reduced the computational complexity of cross attention.

**Questions For The Authors:**

1. The model seems not that computationally efficient compared with other models based on the analysis, though it achieves seemingly better performance, and more experimental results need to be explored based on fair comparison.
2. Do you consider comparing with more previous methods?
3. In this paper, only T5 and BART are used. Do you try testing on larger models now that reducing cost for large models also counts?

**Reasons To Accept:**

1. SRformer achieves better computational complexity and accuracy trade-off through aggregating information across segments with recurrent attention.
2. Experimental results show that the method achieves better performance.

**Reasons To Reject:**

1. Computational reduction is not that surprising compared to other methods.
2. It seems that more fair experiments are needed to demonstrate the effectiveness of the method.

**Reproducibility:**

3: Could reproduce the results with some difficulty. The settings of parameters are underspecified or subjectively determined; the training/evaluation data are not widely available.

**Reviewer Confidence:**

3: Pretty sure, but there's a chance I missed something. Although I have a good feel for this area in general, I did not carefully check the paper's details, e.g., the math, experimental design, or novelty.

---

> ### Author Rebuttal · Authors · 2023-08-29
>
> We thank the reviewer for the comments.
>
> **Regarding the computational reduction**, we improve the computational complexity from O(qkd) to O(qsd + kd^2). In many cases, the length of summary, q, depends on the length of article, k. Then the improvement is from quadratic to linear. The computational reduction increases proportionally with the lengths, as shown in Table A, which is an extension for Table 1 in our paper. When q=128 and k=1024, the overall cost is reduced by 43.8%. When they both increase by 4 times, the improvement will be 85.9%. Moreover, most existing works only reported the computation reduction of attention and ignored the other layers such as feed forward layers in overall cost. We have recorded the actual memory footprints and GPU usage of training different models as shown in Table B. Our method uses the lowest percentages of GPU power usage and memory accessing time, providing 29%/31% improvement over the baseline. Although GPU memory usage of all recurrent transformers is larger than vanilla BART to store weights and gradients, our method uses less memory than TransformerXL and LongT5. Considering these, we believe our improvement is significant. Besides, we have provided experiments on four different datasets in Table 2 in our paper and an ablation study in section 4.3 to validate the effectiveness of the method.
>
> **Table A: Number of computation of cross attention VS length**
> |	q/k	    |    128/1024	|        256/2048	   |     512/4096|
> |---------------|-------------------|--------------------|--------------|
> |Transformer (full attention)|	8.39E+06 |	3.36E+07 |	1.34E+08|
> |SRformer	    |    4.72E+06 |	9.44E+06 |	1.89E+07|
> |reduction	     |   43.8%	|	71.9%	|	85.9%|
>
> **Table B: Memory and power comparisons**
> |			   |     GPU Mem(%)	| GPU mem access time(%)	|Power usage(%)	 |
> |---|---|---|---|
> |BART	(Baseline)	|	49		|	91			|	99|
> |TransformerXL		|	82		|	70			|	89|
> |LongT5				 |       73		|	80			|	90|
> |SRformer - BART		  |      60		|	60			|	80|
> |SRformer - T5		|	        62	|		60		|		81|
>
>
> **Regarding the comparison to existing methods**, we have compared the ROUGE1 scores of the proposed SRformer to Transformer-XL and LongT5 in Fig. 4 (submitted paper). Compared to these existing baselines, only SRfromer achieves comparable results to full attention when small segment sizes are used, and its performance is only slightly affected by the reduction of segment size. Furthermore, we provide a new comparison with another existing work, Longformer [1]. As shown in Table 2 (submitted manuscript), our proposed method with segment size set to 64 achieves 36.04 and 42.99 ROUGE 1 scores on ArXiv dataset when applying to T5 or BART, whereas Longformer with segment size set to 1024 achieves 35.21 using BART and shows a clear trend of decreasing with the segment size (Figure 3 of [1]).
>
> **Regarding testing on larger models**, although larger models have more layers and more heads, the architectures of transformers are similar, and the attention mechanism remains the same. Hence, we expect the proposed method to be valid. The percentage of computational reduction is dependent on the lengths q and k, therefore, it will show similar trends in larger models. Due to resource constraints in an academic setting, it is difficult for us to experiment with training these LLMs. But we thank the reviewer for the suggestion.
>
> [1] Iz Beltagy, Matthew E. Peters, and Arman Cohan. 2020. Longformer: The long-document transformer. CoRR, abs/2004.05150.

---

### Official Review · Reviewer_f7Z1 · 2023-08-03

**Soundness:** 3

**Excitement:**

3: Ambivalent: It has merits (e.g., it reports state-of-the-art results, the idea is nice), but there are key weaknesses (e.g., it describes incremental work), and it can significantly benefit from another round of revision. However, I won't object to accepting it if my co-reviewers champion it.

**Missing References:**

- 'The Devil in Linear Transformer'.  This paper also adopts segmented attention and uses linear attention to fuse information

**Paper Topic And Main Contributions:**

- This paper proposes a new structure for sequence-to-sequence task
- It proposes a segmented attention to reduce the computation complexity of vanilla attention
- It proposes a recurrent module to accumulate the information across the segments.
- The method is validated on summarization benchmarks

**Questions For The Authors:**

- In Fig.2 , why do we still need calculate $(QK^T)V$, can we calculate all attention in the linear form shown in eq 13?

**Reasons To Accept:**

- The paper is well written and easy to follow.
- The motivation is clear, the segment attention can reduce the computation and the recurrent can fuse the global contexts.
- The design of RFA is novel which is a new method to include global information into segments.
- The experiments show the effectiveness of this model on the summarization task.

**Reasons To Reject:**

- The actual memory footprints and speed comparison tables are needed to validate the efficiency. This method should be compared to existing methods such as Transformer-XL, RMT or techniques such as KV-cache since they share similar methodology.
- The novelty of the propose method is limited, it only incremental improvements over the existing block-wise attention and RNN-based attention.
- The summarization task commonly takes long-sequence document as input, why the encoder still uses vanilla attention as it might has large computational overhead?
- With the advent of decoder-only structures, I cannot see the reasons for designing a specific model for summarization task as the decoder-only model can also do this task, Why don't simply integrate this method into the decoder-only structure?

**Reproducibility:**

3: Could reproduce the results with some difficulty. The settings of parameters are underspecified or subjectively determined; the training/evaluation data are not widely available.

**Reviewer Confidence:**

4: Quite sure. I tried to check the important points carefully. It's unlikely, though conceivable, that I missed something that should affect my ratings.

---

> ### Author Rebuttal · Authors · 2023-08-29
>
> Thank you for the feedback and for finding our work well-motivated, novel and effective.
>
> **Comparison to existing works**: Regarding the comparison to existing methods, we have compared the performance of the proposed SRformer to Transformer-XL and LongT5 in Fig. 4 (submitted manuscript). Only our method can achieve results comparable to full attention when small segment sizes are used, and its performance is not affected by the reduction of segment size. We cited RMT in section 2, but since it was a concurrent work, we had not compared to it. Additionally, we provide comparison with another previous work, Longformer [1]. Our proposed method with segment size set to 64 achieves 36.04 and 42.99 ROUGE 1 scores on ArXiv dataset when applying to T5 or BART, whereas Longformer with segment size set to 1024 achieves 35.21 using BART and shows a clear trend of decreasing with the segment size (Figure 3 of [1]). We have recorded the actual memory footprints and GPU usage of training different models as shown in Table B. Our method uses the lowest percentages of GPU power and memory accessing time, providing 29%/31% improvement over the baseline. Although GPU memory usage of all recurrent transformers is larger than vanilla BART to store weights and gradients, our method uses less memory than TransformerXL and LongT5.
>
> **Theoretical improvement:** As mentioned, the proposed method improves the computational complexity from O(qkd) to O(qsd + kd^2). In many cases, the length of summary, q, depends on the length of article, k. Then the improvement is from quadratic to linear. We would like to provide Table A, which is an extension for Table 1 in our paper, to show theoretical computational reduction increases proportionally with the lengths. When q=128 and k=1024, the overall cost is reduced by 43.8%. When they both increase by 4 times, the improvement will be 85.9%.
>
> **Table A: Number of computation of cross attention VS length**
> |	q/k	    |    128/1024	|        256/2048	   |     512/4096|
> |---------------|-------------------|--------------------|--------------|
> |Transformer (full attention)|	8.39E+06 |	3.36E+07 |	1.34E+08|
> |SRformer	    |    4.72E+06 |	9.44E+06 |	1.89E+07|
> |reduction	     |   43.8%	|	71.9%	|	85.9%|
>
> **Table B: Memory and power comparisons**
> |			   |     GPU Mem(%)	| GPU mem access time(%)	|Power usage(%)	 |
> |---|---|---|---|
> |BART	(Baseline)	|	49		|	91			|	99|
> |TransformerXL		|	82		|	70			|	89|
> |LongT5				 |       73		|	80			|	90|
> |SRformer - BART		  |      60		|	60			|	80|
> |SRformer - T5		|	        62	|		60		|		81|
>
> **Block-wise attention and RNN-based attention**: Although we combine block-wise attention and recurrent based attention, we analyze the loss caused by block-wise attention and re-design the recurrent attention to specifically compensate for the lost information. As far as we know, the existing work did not show the complementary relation between local attention and global attention mathematically. Hence, we believe our work gives a novel view of how local attention and global/recurrent attention collaborate.
>
> **Attention in the encoder**: We keep the vanilla attention in the encoder because it runs only once during generation, while the decoder takes steps to generate new tokens. Hence it is acceptable for it to use vanilla attention. Allowing the encoder to access all tokens would make sure the summary captures all information. It is possible to apply the segmentation and recurrence approach to the encoder as well, which will improve efficiency. However, it leads to further information loss before the decoder part resulting in performance degradation, so it is a trade-off.
>
> **Decoder-only models**: It is possible to integrate the proposed method into the decoder-only structures. However, decoder-only models would keep using the original article as inputs at each generation step while concatenating the generated words to its end. Then it becomes difficult to apply segmented attention to the article and summary separately. On the other hand, encoder-decoder models separately process encoded articles and generated summary and explicitly learn input-output relationships. Hence, we think it is necessary to design an efficient encoder-decoder model for summarization task, and our method is more suitable for such a model.
>
> **Response to the question**: In section 4.3 ablation study, we have shown that if we only use recurrent attention in Eq. 13, the ROUGE score would drop by 3. In addition, we have tried linear attention, proposed in paper [2], but it gives ROUGE scores lower than 10. We think the reason might be providing too little information at a time (segment size=1) and a lack of exponential softmax. Hence, we think segmented attention with softmax and recurrent attention are both important and keep them in our design.
>
> We thank the reviewer for referring  to “The Devil in Linear Transformer” [3] as it is highly related to our work. We have noticed that unbounded values could impact the convergence, and our solution is to apply normalization on recurrent (linear) attention and keep using softmax in segmented attention. We will add this paper [3] to our related works.
>
> [1] Iz Beltagy, Matthew E. Peters, and Arman Cohan. 2020. Longformer: The long-document transformer. CoRR, abs/2004.05150.
>
> [2] Angelos Katharopoulos, Apoorv Vyas, Nikolaos Pappas, and François Fleuret. 2020. Transformers are rnns: Fast autoregressive transformers with linear attention. In Proceedings of the 37th International Conference on Machine Learning, ICML’20.
>
> [3] Zhen Qin and XiaoDong Han and Weixuan Sun and Dongxu Li and Lingpeng Kong and Nick Barnes and Yiran Zhong, The Devil in Linear Transformer, 2022, abs/2210.10340.

---

### Meta-Review · Area_Chair_ZFHd · 2023-09-08

**Recommendation:** 4

**Metareview:**

Reviewers agree that the work is moderately sound and moderately exciting. Reviewers like the efficiency and strong experimental results of the novel way to use attention during generation. While some flaws are highlighted there is consensus that the overall method and evaluation is sound. I recommend acceptance to the main conference or findings.

---

### Decision · Program_Chairs · 2023-10-07

**Decision:**

Accept-Findings

**Comment:**

Reviewers agree that the work is moderately sound and moderately exciting. Reviewers like the efficiency and strong experimental results of the novel way to use attention during generation. While some flaws are highlighted there is consensus that the overall method and evaluation is sound. I recommend acceptance to the main conference or findings.